# An Investigation of the Domain Gap in CLIP-Based Person Re-Identification

**DOI:** 10.3390/s25020363

**Published:** 2025-01-09

**Authors:** Andrea Asperti, Leonardo Naldi, Salvatore Fiorilla

**Affiliations:** Department of Informatics—Science and Engineering (DISI), University of Bologna, 40126 Bologna, Italy; leonardo.naldi@studio.unibo.it (L.N.); salvatore.fiorilla@unibo.it (S.F.)

**Keywords:** person re-identification, domain gap, CLIP, deep learning, computer vision

## Abstract

Person re-identification (re-id) is a critical computer vision task aimed at identifying individuals across multiple non-overlapping cameras, with wide-ranging applications in intelligent surveillance systems. Despite recent advances, the domain gap—performance degradation when models encounter unseen datasets—remains a critical challenge. CLIP-based models, leveraging multimodal pre-training, offer potential for mitigating this issue by aligning visual and textual representations. In this study, we provide a comprehensive quantitative analysis of the domain gap in CLIP-based re-id systems across standard benchmarks, including Market-1501, DukeMTMC-reID, MSMT17, and Airport, simulating real-world deployment conditions. We systematically measure the performance of these models in terms of mean average precision (mAP) and Rank-1 accuracy, offering insights into the challenges faced during dataset transitions. Our analysis highlights the specific advantages introduced by CLIP’s visual–textual alignment and evaluates its contribution relative to strong image encoder baselines. Additionally, we evaluate the impact of extending training sets with non-domain-specific data and incorporating random erasing augmentation, achieving an average improvement of +4.3% in mAP and +4.0% in Rank-1 accuracy. Our findings underscore the importance of standardized benchmarks and systematic evaluations for enhancing reproducibility and guiding future research. This work contributes to a deeper understanding of the domain gap in re-id, while highlighting pathways for improving model robustness and generalization in diverse, real-world scenarios.

## 1. Introduction

The goal of person re-identification (re-id) is to match individuals across multiple non-overlapping cameras [1,2,3,4]. In its simplest and most widely studied form, re-id involves matching a given person’s image, referred to as the query, against a set of cross-camera images, referred to as the gallery, as illustrated in Figure 1. Interest in person ReID has grown in both academia and industry, fueled by advances in deep learning, particularly in computer vision, and its applications in intelligent surveillance systems.

Person re-id remains challenging due to the inherent difficulties of a multi-camera setup, including variations in lighting, camera angle, and quality, as well as occlusion. Furthermore, models must rely on appearance-based features, such as clothing, rather than robust biometric features like facial details, which cannot be reliably used [1].

Despite these challenges, research has made significant progress, with most modern benchmarks reporting a Rank-1 score of at least 90% [2,3,5]. This success has been driven by advances in deep learning architectures, particularly ResNet [6,7] and Vision Transformers [5,8,9], along with the development of larger, higher-quality datasets [10,11,12,13,14,15,16].

One of the most significant challenges affecting person re-id models is the domain gap. Even high-performing models experience considerable performance degradation when tested on previously unseen datasets [13,17].

The gap arises from two main factors. First, the task’s complexity makes models highly sensitive to variations in lighting, camera angles, and clothing styles, which are common when data are sourced from different environments than the training dataset. Second, the high cost of collecting and annotating person re-ID data limits the size of available datasets, hindering models’ ability to generalize.

While substantial research has been conducted on mitigating the domain gap through methods like transfer learning, data augmentation, and domain translation/adaptation [13,18,19], there is a notable lack of focus on quantitatively measuring its effects. This oversight hampers the systematic evaluation of proposed solutions and consistent comparison between approaches.

In this article, we provide quantitative measures of the domain gap, with a particular focus on recent CLIP-based models, which hold significant potential for domain adaptation by aligning visual and textual representations. We evaluate the accuracy of different models on standard benchmarks, including the Airport [14], ENTIRe-ID [16], Market-1501 [11], DukeMTMC-reID [12], and CUHK03 [10] datasets as test sets. Specifically, we train the models on a single dataset and measure their performance on the others. This approach allows us to systematically quantify the impact of the domain gap.

We compare two CLIP re-id models with different backbones and their corresponding strong image encoder baselines [20], enabling us to assess the benefits provided by the CLIP component in mitigating the domain gap.

Furthermore, we investigate the feasibility of reducing the domain gap by extending the training set with non-domain-specific data. In particular, we train the best-performing model (ViT CLIP-ReID [20]) on a combined dataset of Market-1501 [11] and Airport [14], both with and without random erasing augmentation [7], and quantify the resulting performance improvements across different datasets. We also observe that adding more datasets to the union consistently results in smaller performance gains, suggesting that strategically selecting challenging datasets based on cross-dataset results is more effective than indiscriminately increasing the training data volume.

The contributions of this paper are thus the following:We quantify the expected gains from extending the training set in a non-domain-specific manner, assessing the actual potential of using large, heterogeneous datasets to bridge the domain gap. This analysis provides a foundation for evaluating the trade-offs between large-scale dataset augmentation and lighter domain adaptation techniques, such as one-shot learning or unsupervised adaptation, offering actionable insights for future research.By comparing the performance of CLIP-ReID models with strong image encoder baselines, we highlight the specific advantages introduced by multimodal pre-training. This analysis underscores the role of CLIP’s visual–textual alignment in mitigating domain discrepancies, offering a clearer understanding of its contribution to person re-id.

## 2. Materials and Methods

Our work is a comparative investigation of models trained on specific datasets and tested on different ones. In this section, we introduce some of the most important person re-id datasets (Section 2.1), including the ones that we used for our experiments, the metrics (Section 2.3), and the class of model objects of our investigation (Section 2.4).

### 2.1. Datasets

Person re-id datasets are usually divided into a train set and a test set, and consist of pedestrian bounding boxes, either hand-crafted or extracted by an object detector [21] and/or an object tracker [22], as can be seen in Figure 2. The identities present in the test set are not present in the training set. In order for a model to be evaluated, the test set will be split into two subsets: the *query* (containing all images that will be used as the query by the model) and the *gallery* (the set that the model will search to find other images depicting the same person as the query). The model will, for each image in the query, search for images in the gallery depicting the same person. In order for this to be feasible, each query identity (that is, each identity that appears at least once in the query set) needs to be present in at least one, but preferably more, gallery images taken from a different camera.

For supervised datasets, each image will usually be annotated with the person’s ID, as well as the camera’s ID. The characteristics of some of the most influential person re-id datasets can be seen in Table 1. In order for datasets to be effective, a wide variety of factors have to be taken into account and represented within the dataset: occlusion, lighting changes, weather changes, varying crowd dynamics, and multiple scenes (e.g., outdoors and indoors) are just some examples.

#### 2.1.1. CUHK03

CUHK03 is a dataset released in 2014 by Li et al. [10] at the Chinese University of Hong Kong. Compared to earlier datasets [23,24,25,26,27], CUHK03 exhibits a much greater magnitude in terms of data volume, with 13,164 bounding boxes and 1360 identities extracted from footage recorded by 6 cameras, with each identity being captured by two cameras and having an average of 4.8 images per identity in each of the two camera views.

#### 2.1.2. Market-1501

Introduced by Zheng et al. [11] in 2015, Market-1501 is a person re-identification dataset consisting of 32,668 bounding boxes representing, as the name implies, 1501 different identities (some examples can be seen in Figure 3). The dataset was created from footage recorded by six cameras in front of a campus market. The dataset also includes junk and distractor images.

Market was also one of the first datasets where each identity could have multiple images under each camera, as opposed to the usual approach of having each identity only captured by 2 cameras [10,28], meaning that both the query and the gallery set may contain images from multiple cameras of the same identity.

#### 2.1.3. DukeMTMC-reID

DukeMTMC-reID (Duke Multi-Target Multi-Camera re-identification) [12] is a dataset released by Ristani et al. [12] in 2016. It was crafted from footage recorded by 8 high-quality cameras placed on Duke University’s campus. It is one of the largest datasets, consisting of hand-crafted bounding boxes [29] extracted from high-quality images. The footage consists of an 85 min recording taken from the time in between lectures (to guarantee high pedestrian activity) for each camera. The cameras record at 1080p and 60 frames per second. Each person was manually tracked by recording the person’s foot point of contact with the ground. Finally, for each of the eight cameras, the first five minutes of footage were reserved for training, and the remaining for testing.

#### 2.1.4. MSMT17

The MSMT17 (**M**ulti-**S**cene **M**ulti-**T**ime) person re-identification dataset, introduced in 2017 by Wei et al. [13], is one of the largest and most challenging supervised person ReID datasets. It was derived from 180 h of video footage captured by 15 cameras installed on a university campus, including 12 cameras covering outdoor scenes under various weather conditions and 3 cameras for indoor environments.

The pedestrian bounding boxes were detected automatically using a Faster R-CNN [30], which improves on the DPM [31] object detector that was the standard in earlier works [10,11] and provides fairly accurate detection. The dataset contains 126,441 bounding boxes spanning 4101 identities, which is a significant improvement over earlier datasets, all of which contained fewer than 2000 identities [10,11,28], as can be seen in Table 1.

#### 2.1.5. Airport

Airport [14] is a person re-identification dataset constructed from footage recorded inside a major airport. The footage was recorded over the course of 12 h in a single day by six cameras placed after the airport’s security checkpoints. Each 12 h video was randomly split into 40 five-minute clips, which were then used to create the dataset.

Airport’s main novelty lies in its camera network and its environment: the cameras used were the airport’s actual security cameras, whose complex orientation and low resolution introduces additional challenges. Furthermore, the inside of an airport contains a greater variety of people and more varying crowd dynamics than a university campus.

The dataset was created during the deployment of a real-time intelligent surveillance system [32], with bounding boxes generated by the ACF framework [33] and tracking performed using the KLT tracker [34] and FAST corner features [35]. It contains 9651 identities, of which 1382 reappear across multiple cameras, while the remaining unpaired identities were included to increase the dataset’s complexity.

#### 2.1.6. ENTIRe-ID

ENTIRe-ID [16] is, to the best of our knowledge, the most recent and most extensive supervised person re-identification dataset, having been released in 2024 and containing more than four million images (as of the time of writing, only the test set has been released).

The ENTIRe-ID datasets contains 4.45 million images spanning 13,540 identities; some example are shown in Figure 4. The source footage was taken from 37 publicly available internet cameras located in four different continents. This massively improves the standard for dataset size: some of the closest supervised datasets in terms of size are the LaST dataset [36], created from movie footage and comprising 228,156 images spanning 10,862 identities, and the MARS dataset [37], which is a video extension of the Market-1501 dataset [11] containing more than one million bounding boxes, but with the same identities and cameras as Market-1501.

Bounding boxes have been created by means of the YOLOv8 object detection model (https://docs.ultralytics.com/it/models/yolov8/ accessed on 5 December 2024), integrated with the ByteTrack [38] tracking algorithm to correctly identify people in consecutive video frames. The face of each person contained in the dataset is blurred, both to preserve the subject’s privacy as well as to push models away from learning and exploiting facial features.

#### 2.1.7. LUPerson

LUPerson [15] is an unlabeled person re-id dataset, containing more than 4 million images spanning over 200 k identities, extracted from over 46 k Youtube videos. The videos were selected by searching for streetview footage recorded in 100 big cities, resulting in roughly 680 to 760 videos per city. The bounding boxes were extracted using YOLO-v5 (https://github.com/ultralytics/yolov5, accessed on 1 December 2024).

LUPerson was conceived as an alternative to ImageNet [39] for unsupervised pre-training which, given the modest size of most supervised datasets, may play an important role in person re-id.

### 2.2. Data Augmentation

The standard pipeline for person re-id generally includes *random erasing augmentation* (REA) [7]. As discussed above, pedestrian bounding boxes often contain some form of occlusion, which contributes to the complexity of the problem [40]. In order to push the model to adapt to partial occlusion, random erasing augmentation was proposed by Zhong et al. [41]. REA works as follows: each image will have a probability of undergoing REA equal to $p_{e}$, and those that do undergo REA have a randomly selected rectangle erased with random pixels, as can be seen in Figure 5.

The effects of REA have been extensively studied by Luo et al. [7]. The study was conducted using ResNet-50 [6] as a backbone and the Market-1501 [11] and the DukeMTMC-reID [12] datasets for training. During training, each image was resized to 256 × 128 pixels, padded with 10 zero-value pixels, and then randomly cropped to a 256 × 128 image.

The images then underwent horizontal flipping with a 50% chance. Finally, each image was encoded in 32-bit float values in the range [0,1], and the RGB channels were normalized. The only training trick used in the baseline was a warmup period for the learning rate lasting 10 epochs. The impact of REA was measured in terms of mAP and CMC Rank-1 (Section 2.3) in two different settings: the *same dataset* setting, where the model was trained and tested on the same dataset, and the *cross-dataset* setting, where the model was trained on one of the two datasets and tested on the other.

Within the same domain, REA was found to improve both mAP and CMC Rank-1 by between 1% and 4%. However, in the cross-dataset setting, applying REA appears to harm the model’s performance by a similar margin. This is conceivably due to the model overfitting the training set in order to compensate for the extra occlusion [7].

### 2.3. Metrics

In order to evaluate trained person re-id models, they are tested on a new set of persons’ images. These new images will be from the same dataset used for training, but will only depict people never seen by the model during training. As said, this testing set will be split into a *query* set and a *gallery*, where each identity present in the query will also be present in a cross-camera image (meaning an image taken by a different camera) of the gallery. The model will then be evaluated on its ability to retrieve, for each query image, images from the gallery depicting the same person. This evaluation is conducted through metrics, and the two most widely used metrics are called *cumulative matching characteristics rank-k* and *mean average precision*.

#### 2.3.1. Cumulative Matching Characteristics

Cumulative matching characteristics rank-k, often shortened to $CMC_{k}$ or *Rank-k*, where *k* will be a positive integer, is, together with mean average precision, one of the two most used metrics to evaluate person re-id models. It represents the probability that a correct sample appears in the highest *k* scored gallery samples [2,5,42].

In mathematical terms, given a dataset D=(T,Q,G) (training set, query set, and gallery set respectively), a model *M*, and a distance function *d*, we have that(1)$$CMC_{k}\left(M,Q,G,d\right)=\frac{1}{\left|Q\right|}\sum_{q\in Q}acc_{k}\left(q,G^{\prime }\right)$$
where $G^{\prime }$ is the output of model *M* on *G* sorted according to the distance *d* from the output of model *M* on the query sample *q* (in layman’s terms, $G^{\prime }$ will be the gallery set sorted according to how close the model currently thinks each gallery image is to query image *q*), and where $acc_{k}$ is the accuracy at the *k* function:(2)$$acc_{k}\left(x,S\right)=\left\{\begin{array}{cc} 1 & \mathrm{if}\;a\;\mathrm{sample}\;\mathrm{matching}\;x\;\mathrm{is}\;\mathrm{in}\;\mathrm{the}\;\text{top-k}\;\mathrm{samples}\;\mathrm{of}\;S\\ 0 & \mathrm{otherwise} \end{array}\right.$$

#### 2.3.2. Mean Average Precision

The $CMC_{k}$ score measures the likelihood that a match for a query image appears within the top *k* results of a ranked list, assuming each query has a single ground truth in the gallery, as seen in older datasets like VIPeR. Modern datasets, such as Market-1501, often have multiple ground truths per query, requiring complementary metrics.

Mean average precision (mAP) is another widely used metric for evaluating person ReID models, calculated as the mean of average precision values across all queries. Average precision considers the precision of the model at various ranks, with precision at rank *k* denoted as $P_{k}$ (or P@k). Below, we recall its formal definition, for the sake of completeness.

Given a sorted scored list $G^{\prime }=\left(g_{1},\cdots ,g_{n}\right)$ of items, each labeled by a ground truth $t_{i}$, a query sample *q* and its ground truth $t_{q}$ (in the case of person re-id, the samples will be images and the ground truths will be the person’s identity) we can define the precision at *k* as the ratio of true positives within the first *k* samples of $G^{\prime }$ and *k*. A sample $g_{i}$ is considered a true positive if its ground truth $t_{i}$ matches $t_{q}$. Thus, the number of true positives within the first *k* samples for query *q* can be written as follows:(3)$$TP_{k}\left(q\right)=\left|\left\{g_{i}\in \left(g_{1},\cdots ,g_{k}\right):t_{i}=t_{q}\right\}\right|$$

The precision $P_{k}\left(q\right)$ at *k* for query sample *q* is(4)$$P_{k}\left(q\right)=\frac{1}{k}TP_{k}\left(q\right)$$

We also need to define the relevance-at-k function:(5)$$rel_{k}\left(q\right)=\left\{\begin{array}{cc} 1 & \mathrm{if}\;t_{k}=t_{q}\\ 0 & \mathrm{else} \end{array}\right.$$

Then, the average precision AP(q) for a given query *q* is(6)$$AP\left(q\right)=\frac{1}{TP_{n}\left(q\right)}\sum_{k=1}^{n}P_{k}\left(q\right)\cdot rel_{k}\left(q\right)$$

Mean average precision is simply the mean of the AP function over all query samples:(7)$$mAP=\frac{1}{\left|Q\right|}\sum_{q\in Q}AP\left(q\right)$$

### 2.4. Models

We focused on the CLIP-ReID framework [20], which is the adaptation of the CLIP framework [43] by OpenAI to person re-id. This is due to a number of factors.

First and foremost, person re-id is conceptually similar to zero-shot classification [43]: models will be evaluated on (and faced with once they are deployed in a real-world scenario) a set of identities that were not present in the training set. If the identity of each person is considered to be a classification label, then the person re-id task becomes the classification of unseen labels (zero-shot classification [43]) taken from an open-ended set of labels, which CLIP excels at.

Additionally, CLIP models are able to learn fine-grained visual features which, together with the nearly unlimited scope of natural language that CLIP is able to exploit, allow it to easily adapt to other tasks and, more importantly, domains. This ability to adapt to new domains should, hopefully, transfer to person re-id and help bridge the domain gap.

Another reason for selecting CLIP-based models is their ability to learn rich and fine-grained visual representations, which make them particularly suitable for person re-id, where subtle differences in appearance, pose, and context can be crucial for accurate identification.

The CLIP-ReID models also obtained the highest scores at the time of writing on the MSMT17 dataset [13] (one of the most challenging supervised person re-id datasets) on the Papers with Code website (https://paperswithcode.com/, accessed on 7 December 2024).

Together with the CLIP-ReID models, we used the corresponding baseline models (which consist of CLIP’s image encoder [20]) to measure the impact of the CLIP-ReID framework in the context of the domain gap.

In this section, we introduce the CLIP framework (Section 2.4.1), the baseline models (Section 2.4.2), and the CLIP-ReID framework (Section 2.4.3).

#### 2.4.1. CLIP

CLIP (Contrastive Language–Image Pre-training) [43] is a framework created by OpenAI, which aims at associating images and natural text. This allows the models to exploit natural language features, which have a nearly unlimited scope.

Specifically, given *N* images and *N* prompts, CLIP will determine which of the N×N possible (image,text) pairings actually occurred. This is achieved by jointly training a text encoder (a transformer [44]) and an image encoder (either a ResNet [6] or a Vision Transformer [8]) on a shared embedding space, using a dataset of roughly 400 million (image,text) pairs extracted from the internet.

In practice, CLIP learns a multimodal embedding space, which is shared by both the image and the text features, and it maximizes the cosine similarity of the true *N* pairs of the batch while minimizing it for the remaining $N^{2}-N$ negative pairs. In more mathematical terms, given a batch of size $B\in Z^{+}$ composed of $\left\{img_{1},\cdots ,img_{B}\right\}$ images and $\left\{text_{1},\cdots ,text_{B}\right\}$ pieces of text, the similarity between image $img_{i}$ and text $text_{i}$ is computed as follows [20,43]:(8)$$S\left(V_{i},T_{i}\right)=g_{V}\left(I\left(x_{i}\right)\right)\cdot g_{T}\left(T\left(text_{i}\right)\right)$$
where:I(·) and T(·) are the functions computed by the image and text encoders, respectively.$g_{V}\left(\cdot \right)$ and $g_{T}\left(\cdot \right)$ are the linear layers that project the given (image and text, respectively) embedding into the shared embedding space.
These similarities are optimized using two contrastive losses, called image-to-text and text-to-image, respectively [20,43]:(9)$$L_{i2t}\left(i\right)=-\log\frac{\exp\left(S\left(V_{i},T_{i}\right)\right)}{\sum_{a=1}^{B}\exp\left(S\left(V_{i},T_{a}\right)\right)}$$(10)$$L_{t2i}\left(i\right)=-\log\frac{\exp\left(S\left(V_{i},T_{i}\right)\right)}{\sum_{a=1}^{B}\exp\left(S\left(V_{a},T_{i}\right)\right)}$$

In these two losses, the numerator uses the similarity of the image,text pair that actually matches, and thus has to be maximized, whereas the denominator uses the similarities between the other, non-matching, pairs, which will be minimized.

The immense scope of natural language allows CLIP to be easily adapted to other tasks, without the need for further training (which is referred to as zero-shot setting [43]). One important example of this is classification: CLIP can easily be adapted to classification by crafting a prompt for each label (e.g., “A photo of a {label}”.), as can be seen in Figure 6.

In the context of classification, Zero-Shot CLIP has obtained results competitive with a fully trained baseline, outperforming it on 16 out of 27 datasets [43].

#### 2.4.2. Baseline Models

We used Li et al.’s baseline [20] architecture, which consists of CLIP’s image encoder. Specifically, we used ResNet-50 and ViT-B/16:For the **ResNet** model, the standard ResNet-50 architecture was enhanced by using the ResNet-D architecture [43,45], the anti-aliasied rect-2 BlurPool by Zhang [46], and by replacing the global average pooling layer with an *attention pooling* layer (meaning a single layer of multi-head QKV attention, similar to a transformer mechanism, where the query is based on the globally average-pooled representation of the image) [43].For the **ViT** model, CLIP’s implementation follows the original [8] closely, with the only addition of a normalization layer to the final combination of patch and position embeddings [43].

The baseline models, pre-trained with CLIP and fine-tuned using standard person re-id losses (triplet loss and ID loss, Section 2.4.3), as seen in Figure 7, already show superior performance compared to other common baselines pre-trained on ImageNet (such as the Bag of Tricks Strong Baseline [7] for CNN-based models and the TransReID baseline [47] for ViT-based models) [20]. This suggests that CLIP’s pre-training imparts more robust and transferable features for person re-id than ImageNet, even without the additional enhancements of the full CLIP-ReID framework.

#### 2.4.3. CLIP-ReID

Person re-identification is conceptually similar to zero-shot classification: if the identities of the people present in any given supervised dataset are seen as classification labels, then person re-id can be interpreted to be the classification of never-before-seen identities (meaning, as said, a zero-shot classification problem). This similarity implies that CLIP, which excels at zero-shot classification, should adapt well to person re-id.

While exploiting CLIP seems like a promising premise, in practice it presents some challenges. Namely, the person labels used in person re-id are simple numerical values that lack any concrete natural language description. This was first explored by Li et al. [20]. The lack of exploitable meaning in the IDs was overcome by dividing the training in two stages:During the first stage, the two encoders are kept frozen (meaning their weights do not get altered), and a set of learnable tokens ${\left[X\right]}_{1},\cdots ,{\left[X\right]}_{M}$ is optimized by using it to construct a textual prompt “A photo of a ${\left[X\right]}_{1}\cdots {\left[X\right]}_{M}$ person”, as can be seen in Figure 8. This effectively overcomes the need for a concrete text label. The idea of using learnable tokens to create CLIP’s textual prompt was first introduced by Zhou et al. [48], who sought to overcome the limitations of manual prompt engineering. The loss function used during this stage is the sum of the two clip losses of Equations (Equation 9) and (Equation 10).During the second stage, the learnable token’s weights are frozen and the image encoder’s weights are unfrozen, as can be seen in Figure 8. The training loss function used in this stage is a combination of ID loss, triplet loss [7,20], and an adaptation of the $L_{i2t}$ loss of Equation (Equation 9).

Each image $img_{i}$ in the training dataset is expected to have a person ID (*pid* for short) associated with it, $y_{i}$. This *pid* is used to compute the textual prompt (specifically the tokens) associated with image $img_{i}$ as “An image of a ${\left[X\right]}_{1}\cdots {\left[X\right]}_{M}$ person”.

In Table 2, we provide the results of the CLIP-ReID framework computed by the authors on some of the datasets considered for our experiments. We mostly report them for the sake of completeness and for facilitating a more direct comparison.

As is clear from the table, the ClipReID framework appears to increase the model’s performance with respect to the baseline across all datasets.

The scores can be improved with the embedding of side information (e.g., to make the model aware of the camera) and by using overlapping patches [20]. Re-ranking [2] (the practice of reordering the ranked list produced by the model, usually by searching the list using the first image as a query) can further improve the scores, which become 86.7 mAP and 91.1 Rank-1 on the MSMT17 dataset [13] (the top scores on the Papers with Code website (https://paperswithcode.com/, accessed on 2 December 2024).

## 3. Results

In order to investigate the applicability of person re-id models in the real world, we sought to evaluate their adaptability to new scenarios which could differ widely from the conditions presented in any training dataset. Unsurprisingly, the great amount of variation that person re-id models have to account for cannot be fully captured by a single dataset [17] (due to human appearance itself, the way this appearance is recorded, and other environmental factors), and it has been found that models trained and evaluated on a dataset will perform poorly on data that differ noticeably from the training dataset, which is referred to as the domain gap.

This domain gap is a known challenge in person re-id [13,17] and other computer vision tasks [43], and is a fundamental obstacle to the deployment of a real-world re-identification system. Despite there being substantial research on mitigating it, quantitative measures of its effects are not common. We therefore endeavored to accurately measure its effects on the performance of the CLIP-ReID models [20], and perform some small experiments attempting to mitigate them. Attempts towards bridging this gap have already been made, usually in the form of domain adaptation [13,17] or achieving more generalized features, and although some success has been found, this challenge is yet to be fully overcome.

### 3.1. Cross-Dataset Tests

In order to assess the models’ performance on new data, we took the CLIP-ReID models (including the baselines) [20] trained on the Market-1501 [11] and DukeMTMC-reID [12] datasets, and measured their performance on different datasets. These tests are defined as cross-dataset tests [18,19] and are performed using the following datasets: CUHK03 [10], Airport [14], the test set of ENTIRe-ID [16], Market-1501 [11] (for models trained on DukeMTMC-reID), and DukeMTMC-reID [12] (for models trained on Market-1501). The results of these tests are shown in Table 3 and Table 4.

The checkpoints trained on DukeMTMC-reID [12] obtained similar cross-dataset scores, as shown in Table 4, but a bit lower than those obtained by their Market-1501-trained counterparts. This leads us to believe that Market is, despite its smaller size, more complex than Duke and thus models learn slightly more general features from it.

When compared to the scores obtained by the same model in the traditional setting (Table 2), the degradation in performance is obvious. Across all models and all tests we performed, Airport [14] is the dataset where the lowest scores were obtained. This is probably due to the difference in scene between airport and most other datasets: both Market-1501 and DukeMTMC-reID were created from footage taken in a university campus [11,12], with the cameras being set up by researchers, whereas Airport was created using the surveillance cameras of an actual airport. This leads to a wide difference in camera angles, image quality, activities, clothing, etc., making Airport the most challenging dataset for models trained on Market-1501 [11] and DukeMTMC-reID [12].

### 3.2. Training on the Union of Datasets

In order to help close the domain gap, we re-trained the best-performing model (ViT CLIP-ReID [20]) on the union of Market-1501 [11] (the original training dataset, which yielded the best results in a cross-dataset setting) and Airport [14] (the dataset that proved to be the most challenging in a cross-dataset setting) in the hopes that more general re-id features could be learned from the resulting union. We first re-trained the ViT CLIPRe-ID model, using the same training algorithm and settings used by Li et al. [20], and repeated the cross-dataset tests with the remaining datasets. The results can be seen in Table 5.

The idea of training a model on the union of two or more datasets is not new. For example, Marchwica et al. [50] sought to achieve scene-independent re-id by training on a larger amount of data, obtained by merging between two and six datasets. It is worth mentioning that, following the work of Xiao et al. [51], they balanced the union of datasets by keeping ten images per person, making sure to keep images from all cameras the person appeared in (except for when the person appeared in more than ten cameras, in which case one image per camera was kept). This is carried out to account for the fact that different datasets might have more or fewer images per person, which could lead to overfitting one of the datasets [50]. However, when merging only two datasets, this balancing can lead to a great reduction in the number of images used, so we opted to not balance the union.

Additionally, Luo et al. [7] found that REA (Section 2.2), which is used in the CLIP-ReID training process [20], harms model performance in a cross-dataset setting, and therefore we repeated the above training process and testing procedures without it; the results can be seen in Table 6.

As can be seen in Table 6, the mAP score over all datasets improved by between 4.1 and 4.7 and, similarly, the CMC Rank-1 improved by between 3.4 and 4.7 for the model trained without REA. Meanwhile, adding Airport [14] to the dataset did not considerably hinder performance on the original Market-1501 [11] dataset, which dropped by 2.3 (mAP) and 0.9 (Rank-1). Naturally, the performance on the Airport [14] dataset was greatly improved. It is worth noting that, in order to exploit the differences between Market-1501 [11] and Airport [14] as much as possible, we created the training set by adding every Airport image annotated with a valid and reappearing person ID to Market’s training set, so the scores obtained by the model on Airport [14] should not be given much weight, as they were added for the sake of completeness. Finally, as was expected, the cross-dataset scores experienced a slight improvement when removing random erasing augmentation from the training process, except on the ENTIRe-ID [16] dataset, where the model trained with REA obtained, surprisingly enough, higher cross-dataset scores. This is possibly due to the fact that the people represented in the ENTIRe-ID dataset [16] have their faces blurred, as can be seen in Figure 4, which is a form of artificial occlusion not too dissimilar from the one produced by REA, and thus it is possible that the models trained using it were better equipped to handle the ENTIRe-ID dataset [16].

### 3.3. Additional Experiments

To further investigate the factors contributing to performance improvements, we extended our experiments by including additional datasets in the training process. This allowed us to assess whether the observed performance gains were driven solely by the increase in the quantity of training data, or if strategically selecting datasets based on cross-dataset scores to identify and include the most challenging or “distant” datasets, as was performed in Section 3.2, leads to more effective performance improvements compared to simply adding more data indiscriminately.

To this end, we created two different training datasets, starting from the union of Market-1501 [11] and Airport [14] described in Section 3.2:In the first union, we added the training set from DukeMTMC-reID [12].In the second union, we added the training set from CUHK03 [10] and the test set from ENTIRe-ID [16].

As was carried out in Section 3.2, we trained the ViT CLIP-ReID model [20], both with and without REA (Section 2.2), and repeated the cross-dataset tests for with the remaining datasets. The results are presented in Table 7 (for the model trained on the first union) and Table 8 (for the model trained on the second union).

For both unions, the additional data seem to improve the cross-dataset scores; however, the improvements are reduced when compared to the ones obtained by simply adding Airport (Table 6). In particular, adding DukeMTMC-reID [12] only improved the performance on CUHK03 [10] by 0.4% (mAP) and 0.4% (Rank-1). The additional data had the greatest impact on the cross-dataset scores of the ENTIRe-ID dataset [16], where once again REA seems to help the cross-dataset scores rather than hinder them. When using REA, adding DukeMTMC-reID [12] to the training set improved the performance by 4.4% (mAP) and 4.2% (Rank-1); without REA, it improved by 4.6% (mAP) and 4.1% (Rank-1).

Similarly, adding CUHK03 [10] and the test set of ENTIRe-ID [16] only improved the performance on DukeMTMC [12] by 2.3% (mAP) and 0.8% (Rank-1).

Adding more datasets to the union consistently resulted in a smaller improvement in performance (the only close score improvement being the one obtained on ENTIRe-ID [16] without REA). These findings suggest that leveraging cross-dataset evaluations to strategically identify and incorporate the most challenging dataset is more effective for improving model performance than merely increasing the quantity of training data without considering dataset relevance.

## 4. Conclusions

Despite the rapid advancements in the field of person re-id [2,3,4,17], deploying these models in real-world settings with the same high accuracy seen in benchmark environments remains a challenging goal. The domain gap, or the performance drop when models are applied to data that differ significantly from their training datasets, continues to be a primary obstacle.

In this study, we experimentally measured the effects of the domain gap on state-of-the-art models [20] using various benchmarking datasets, including Airport [14], Market-1501 [11], and DukeMTMC-reID [12]. Our results demonstrate that CLIP-ReID [20] models, while excelling in controlled benchmark scenarios, experience significant performance degradation when confronted with datasets that differ substantially from their training data. This finding underscores the limitations of current models in adapting to diverse, real-world environments.

To address this challenge, we experimented with re-training the best-performing model on a combined dataset of the two most distinct benchmarks in our study. This approach aimed to reduce the domain gap without requiring a dramatic increase in labeled data. Our re-training method achieved an average improvement of +4.3% in mean average precision (mAP) and +4.0% in cumulative matching characteristic (CMC) Rank-1 accuracy, showing a modest yet meaningful enhancement in model robustness across different domains.

While these results represent a positive step, they highlight the ongoing need for innovative domain adaptation strategies to make person re-id models truly adaptable to varied real-world settings. Future work should continue exploring techniques that enable models to generalize across diverse environments, potentially through few-shot learning or incorporating more varied data sources. With recent advancements in generative AI, unsupervised domain adaptation—leveraging the generation of synthetic variants of specific identities—could enhance model generalization without extensive labeled data. Utilizing such generative approaches for domain translation [52] and feature alignment [53] is a promising path toward making re-id models more resilient to environmental variability, ultimately supporting more reliable deployment across a wide range of operational scenarios.

## Figures and Tables

**Figure 1 sensors-25-00363-f001:**
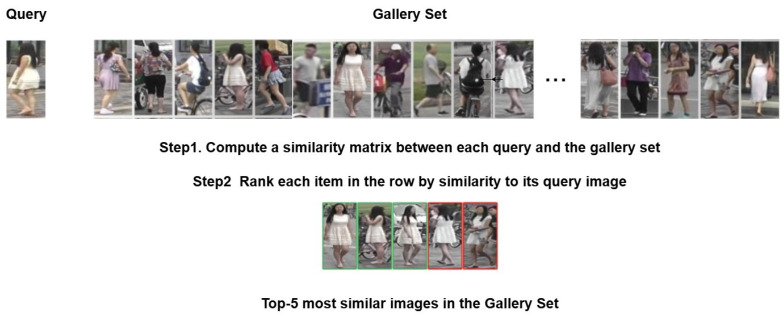
Person re-identification task: the query image of a woman is matched against the gallery set produced by *n* independent cameras. The output is an ordered list of gallery images that the model predicts depict the same person as the query.

**Figure 2 sensors-25-00363-f002:**
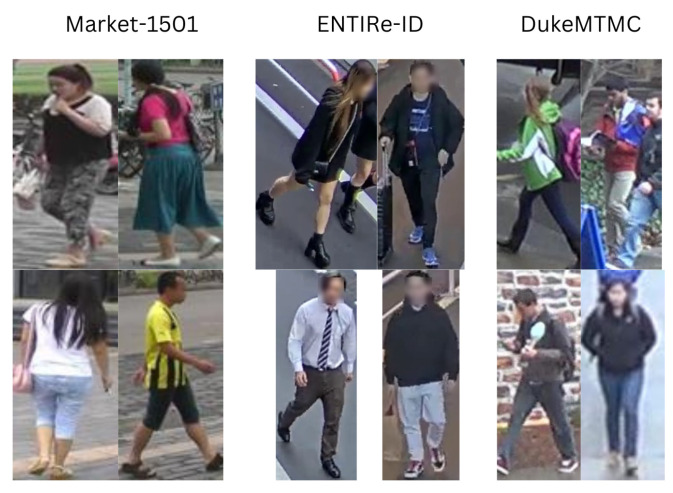
Examples of images from three different person re-id datasets: Market-1501 [11], ENTIRe-ID [16], and DukeMTMC-reID [12].

**Figure 3 sensors-25-00363-f003:**
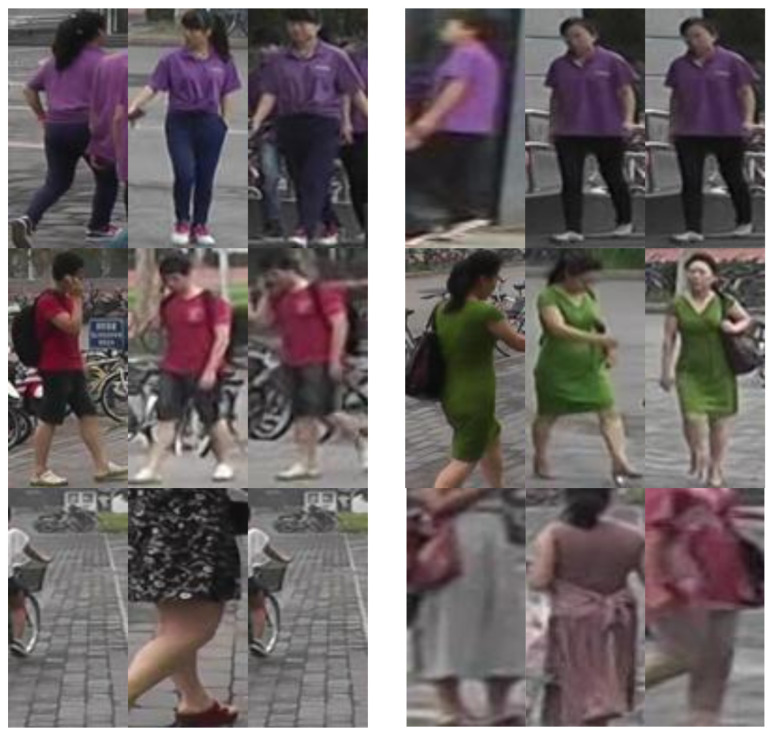
Some examples of Market-1501 images: the top row contains two distinct identities that share a very similar appearance, the middle row consists of two distinct identities that are easily told apart, and the bottom row consists of distractor and junk images.

**Figure 4 sensors-25-00363-f004:**
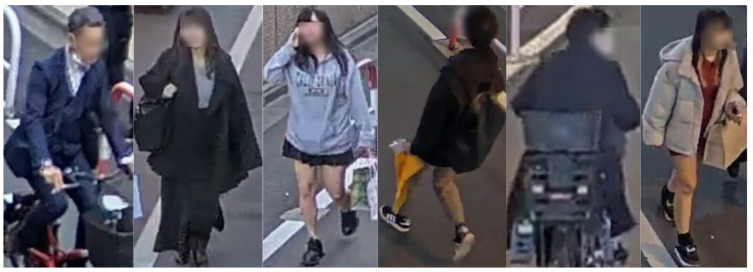
Examples of images from the ENTIRe-ID dataset.

**Figure 5 sensors-25-00363-f005:**
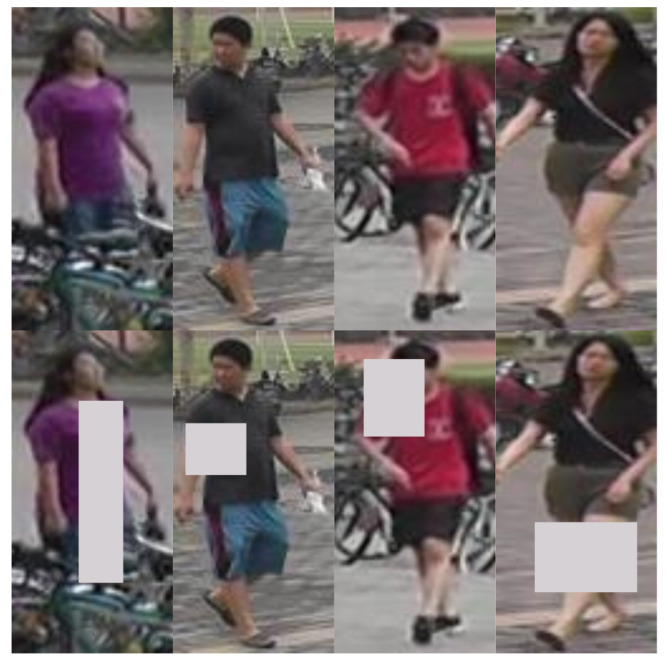
Examples of images undergoing REA.

**Figure 6 sensors-25-00363-f006:**
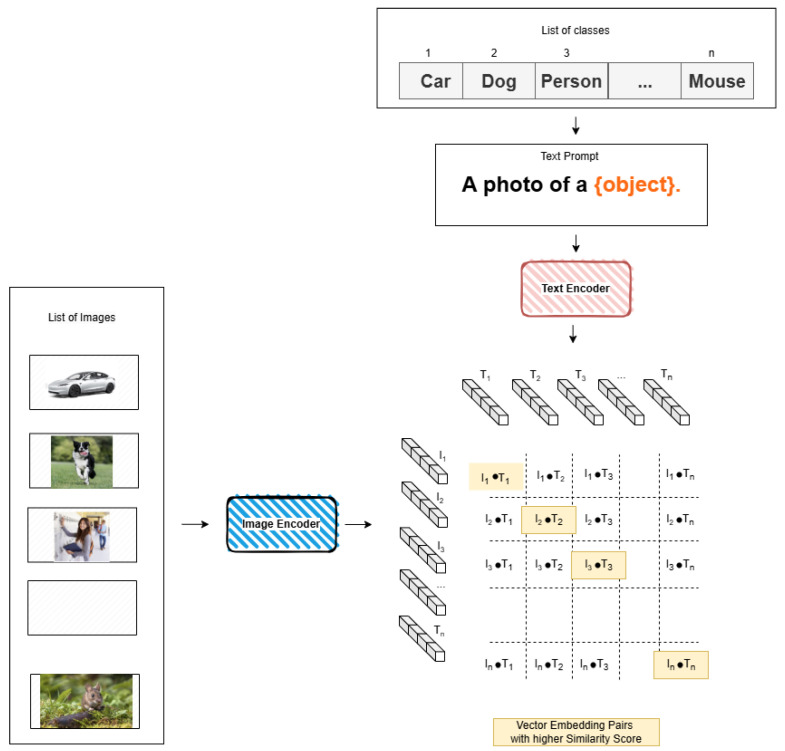
An example of CLIP being adapted to image classification.

**Figure 7 sensors-25-00363-f007:**
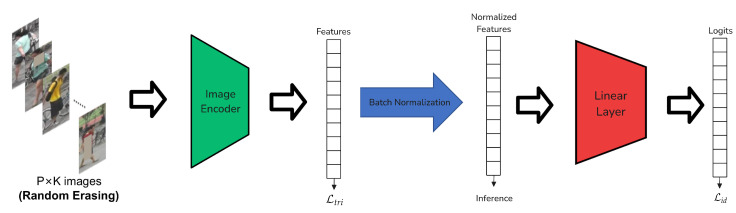
Architecture outline for the baseline models. $L_{tri}$ is a triplet loss and $L_{id}$ is a classification loss for the different identities.

**Figure 8 sensors-25-00363-f008:**
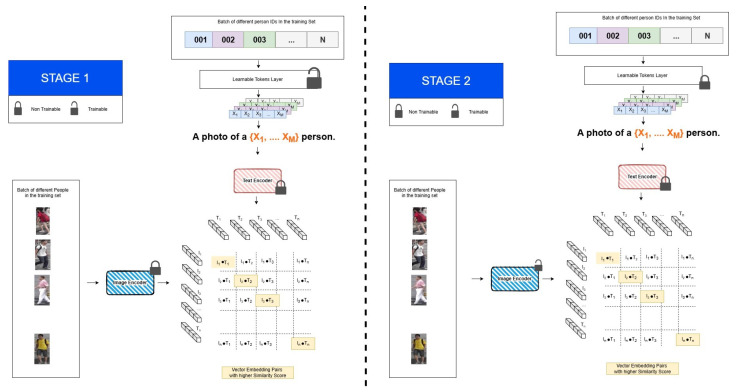
The CLIP-ReID framework [20]. The framework trains the model in two stages. In the first stage, embeddings for tokens representing each ID are learned and concatenated with the text prompt embedding, “a photo of a […] person”. This combined input is fed into the text transformer to calculate the loss. In the second stage, the visual encoder is trained, guided by embeddings from the frozen text encoder.

**Table 1 sensors-25-00363-t001:** Characteristics of some of the person re-id datasets.

Dataset	Year	Identities	Images	Cameras
CUHK03 [10]	2014	1360	13,164	6
Market-1501 [11]	2015	1501	32,668	6
DukeMTMC-reID [12]	2016	1812	36,441	8
MSMT17 [13]	2017	4101	126,441	15
Airport [14]	2018	9651	39,902	6
ENTIRe-ID [16]	2024	13,540	4.45 M	37
ENTIRe-ID (Testing Set ^1^)	2024	2741	13,415	37
LUPerson ^U^ [15]	2021	>200 k	4.18 M	46,260

^1^ As of the time of writing, only the test set has been released. ^U^ Unlabled datasets.

**Table 2 sensors-25-00363-t002:** Scores obtained by the CLIP-ReID framework [20] and by the corresponding baseline architecture.

Backbone	Method	MSMT17 [13]		Market-1501 [11]		DukeMTMC [12]		Occluded Duke [49]	
		mAP	Rank-1	mAP	Rank-1	mAP	Rank-1	mAP	Rank-1
ResNet-50	Baseline	60.7	82.1	88.1	94.7	79.3	88.6	47.4	54.2
	CLIP-ReID	63.0	84.4	89.8	95.7	80.7	90.0	53.5	61.0
ViT-B/16	Baseline	66.1	84.4	86.4	93.3	80.0	88.8	53.5	60.8
	CLIP-ReID	73.4	88.7	89.6	95.5	82.5	90.0	59.5	67.1

**Table 3 sensors-25-00363-t003:** Cross-dataset scores obtained by the CLIP-ReID models [20] and their baseline counterparts, all trained on **Market-1501** [11].

Backbone	Method	Airport [14]		ENTIRe-ID [16]		DukeMTMC [12]		CUHK03 [10]	
		mAP	Rank-1	mAP	Rank-1	mAP	Rank-1	mAP	Rank-1
ResNet-50	Baseline	3.8	5.1	11.9	11.7	17.3	30.8	8.7	8.6
	CLIP-ReID	4.8	6.0	13.3	12.8	21.0	36.8	8.8	8.9
ViT-B/16	Baseline	16.0	18.4	29.3	29.2	44.8	64.3	34.9	36.9
	CLIP-ReID	20.0	22.4	38.9	38.4	50.2	68.9	38.6	40.4

**Table 4 sensors-25-00363-t004:** Cross-dataset scores obtained by the CLIP-ReID models [20] and their baseline counterparts, all trained on **DukeMTMC-reID** [12].

Backbone	Method	Airport [14]		ENTIRe-ID [16]		Market-1501 [11]		CUHK03 [10]	
		mAP	Rank-1	mAP	Rank-1	mAP	Rank-1	mAP	Rank-1
ResNet-50	Baseline	4.5	6.3	19.7	19.4	21.5	47.5	6.1	5.9
	CLIP-ReID	5.3	7.2	21.2	20.4	24.6	52.2	5.7	5.0
ViT-B/16	Baseline	10.0	12.9	33.6	32.9	37.4	62.2	19.2	20.4
	CLIP-ReID	14.3	17.4	40.2	40.3	43.4	70.7	25.9	27.6

**Table 5 sensors-25-00363-t005:** Cross-dataset test result for the ViT CLIP-ReID model [20] trained on the union of Market-1501 [11] and Airport [14]. The difference in scores (the Δ columns) are given with respect to the corresponding cross-dataset scores obtained by the same model trained on the Market-1501 dataset alone, as shown in Table 3. For the sake of completeness, the scores obtained on Market-1501 [11] and Airport [14] are also provided.

Dataset	mAP	ΔmAP	Rank-1	ΔRank-1
ENTIRe-ID [16]	46.2	+7.3	45.9	+7.5
DukeMTMC [12]	53.5	+3.3	71.6	+2.7
CUHK03 [10]	41.8	+3.2	42.6	+2.2
Market-1501 [11]	89.0	−0.6	95.0	−0.4
Airport [14]	64.0	+44.0	56.9	+34.5

**Table 6 sensors-25-00363-t006:** Cross-dataset test result for the ViT CLIP-ReID model [20] trained on the union of Market-1501 [11] and Airport [14] **without using REA** (Section 2.2). The difference in scores (the Δ columns) are given with respect to the corresponding cross-dataset scores obtained by the same model trained on the Market-1501 dataset alone, as shown in Table 3. For the sake of completeness, the scores obtained on Market-1501 [11] and Airport [14] are also provided.

Test Set	mAP	ΔmAP	Rank-1	ΔRank-1
ENTIRe-ID [16]	43.6	+4.7	43.1	+4.7
DukeMTMC [12]	54.3	+4.1	72.3	+3.4
CUHK03 [10]	42.7	+4.1	44.3	+3.9
Market-1501 [11]	87.3	−2.3	94.6	−0.9
Airport [14]	69.4	+49.4	63.1	+40.7

**Table 7 sensors-25-00363-t007:** Cross-dataset test result for the ViT CLIP-ReID model [20] trained on the union of Market-1501 [11], Airport [14], and DukeMTMC-reID [12].

Rea?	ENTIRe-ID [16]		CUHK03 [10]	
	mAP	Rank-1	mAP	Rank-1
Yes	50.6	50.1	41.6	43.1
No	48.2	47.2	43.1	44.7

**Table 8 sensors-25-00363-t008:** Cross-dataset test result for the ViT CLIP-ReID model [20] trained on the union of Market-1501 [11], Airport [14], ENTIRe-ID [16], and CUHK03 [10].

Rea?	DukeMTMC-reID [12]	
	mAP	Rank-1
Yes	54.5	70.7
No	56.6	73.1

## Data Availability

The ALERT Airport Re-Identification Dataset used in the research related to this publication was generated and provided by ALERT (Awareness and Localization of Explosives-Related Threats), a Department of Homeland Security Center of Excellence (COE). The views and conclusions contained in this document are those of the authors and should not be interpreted as necessarily representing the official policies, either expressed or implied, of the U.S. Department of Homeland Security. All other datasets are public domain.

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
