# Peer review of "An Investigation of the Domain Gap in CLIP-Based Person Re-Identification"

_sensors, 2025, doi:10.3390/s25020363_

Round 1
Reviewer 1 Report
Comments and Suggestions for Authors
To investigate the domain gap in CLIP-based person re-identification, this paper proposed a comprehensive quantitative analysis of the domain gap in CLIP-based re-id systems across standard benchmarks. Additionally, this paper evaluated the impact of extending training sets with non-domain-specific data and incorporating random erasing augmentation. However, I have some concerns about this work.
1. Lack of contribution. This paper experimented with an existing method [1] in cross-dataset setting, combining two datasets as training set and verified the impact of using random erasing augmentation. There is no improvement in the method, and multi-dataset combined training and random erasing augmentation are both common methods for re-id, so this paper lacks contribution.
2. The content is redundant. The paper uses a lot of space to introduce the datasets, evaluation metrics, basic concepts of CLIP, and existing work [1]. It is recommended that the authors reduce the length of these contents and focus on highlighting their own contributions.
3. Authors should not use original figures from other papers, such as Figure 6 and Figure 8.
4. Please note the table format specifications, such as Table 2, Table 3 and Table 4.
[1] Li, Siyuan, Li Sun, and Qingli Li. "CLIP-ReID: exploiting vision-language model for image re-identification without concrete text labels." Proceedings of the AAAI Conference on Artificial Intelligence. Vol. 37. No. 1. 2023.
Author Response
Please, check the attached file.

Reviewer 2 Report
Comments and Suggestions for Authors
The manuscript aims to investigate the domain gap in CLIP-Based person re-identification. A comprehensive introduction of the background knowledge is given including the commonly used re-id datasets and the state-of-the-art CLIP-based methods. While it provides a good start for readers who are not familiar with the topic of re-id, the contribution of this study is limited. Details of my comments are as follows.
-- There are a series of works focusing on cross-dataset re-id tasks [1-2] but the authors fail to mention and compare with them. The authors are supposed to do a thorough literature review and apply these techniques to the investigated models in the experiments.
-- In the training experiments on the union of datasets, why only combine two of the datasets? Would it be more beneficial to combine more datasets for training?
-- The authors only investigated the REA as the data augmentation technique, how about other more advanced ones such as pose-guided image synthesis [3]?
[1] Cross-Dataset Person Re-Identification via Unsupervised Pose Disentanglement and Adaptation.
[2] Structure alignment of attributes and visual features for cross-dataset person re-identification
[3] Contrastive Clothing and Pose Generation for Cloth-Changing Person Re-Identification
Author Response
Please, check the attached file

Round 2
Reviewer 1 Report
Comments and Suggestions for Authors
Thanks for the author's reply, but the author's reply to the paper's contribution did not reduce my worries, so I choose to keep the score unchanged.
Reviewer 2 Report
Comments and Suggestions for Authors
The authors have addressed my concerns in the revised manuscript.